# Imaging Flow Cytometry Demonstrates Physiological and Morphological Diversity within Treated Probiotic Bacteria Groups

**DOI:** 10.3390/ijms24076841

**Published:** 2023-04-06

**Authors:** Jakub Kiepś, Wojciech Juzwa, Radosław Dembczyński

**Affiliations:** Department of Biotechnology and Food Microbiology, Poznan University of Life Sciences, Wojska Polskiego 48, 60-627 Poznań, Poland; wojciech.juzwa@up.poznan.pl

**Keywords:** viability, fluid bed drying, lactic acid bacteria, stress factors, quality control, rapid assessment

## Abstract

Probiotic bacteria can be introduced to stresses during the culturing phase as an alternative to the use of protectants and coating substances during drying. Accurate enumeration of the bacterial count in a probiotic formulation can be provided using imaging flow cytometry (IFC). IFC overcomes the weak points of conventional, commonly used flow cytometry by combining its statistical power with the imaging content of microscopy in one system. Traditional flow cytometers only collect the fluorescence signal intensities, while IFC provides many more steps as it correlates the data on the measured parameters of fluorescence light with digitally processed images of the analyzed cells. As an alternative to standard methods (plate cell counts and traditional flow cytometry) IFC provides additional insight into the physiology and morphology of the cell. The use of complementary dyes (RedoxSensorTM Green and propidium iodide) allows for the designation of groups based on their metabolic activity and membrane damage. Additionally, cell sorting is incorporated to assess each group in terms of growth on different media (MRS-Agar and MRS broth). Results show that the groups with intermediate metabolic activity and some degree of cellular damage correspond with the description of viable but nonculturable cells.

## 1. Introduction

Probiotic bacteria, which mainly include lactic acid bacteria (LAB), confer various health benefits. Among them, researchers often describe the stimulation of the immune system providing improvement in immunoinflammatory disorders and allergies, reduced risk of colon cancer and modulation of intestinal microbiota [1]. Multiple health benefits are the main reason for the constant interest in probiotic preparations, their properties and methods of their production, as well as tools for their analysis.

Traditionally, probiotic bacteria were supplied in frozen or liquid form, however these have been superseded by the introduction of more convenient dried formulations. Most lactic acid starter cultures and probiotics are prepared in the form of freeze-dried preparations, but alternative drying techniques such as spray drying, fluid bed drying, vacuum drying, or mixed drying systems are gaining in popularity and are being constantly improved [2]. Their main advantages over freeze-drying are high cost efficiency and lower process duration; on the other hand, methods such as spray drying and fluid bed drying introduce additional stress factors including thermal inactivation, and shearing forces occurring during the atomization of the cell suspension. In both cases, fluid bed drying provides lower air temperature and atomizing pressure, which makes it better suited to use with thermolabile and susceptible materials [3]. The obtained probiotic preparations are secured in many ways to ensure the best possible shelf-life and gastrointestinal survival. Fluid bed drying allows the coating of the preparation with protective and coating substances during the process to ensure their enhanced viability [4]. It can be also used to coat preparations obtained by different methods and can be applied to obtain controlled release of the final product [5]. Protective substances and selected carriers (trehalose, maltose, maltodextrin, lactose, oligofructose, inulin) are also used during spray drying as thermal protectants and to lower the water activity in the preparation for a longer shelf-life [6,7]. An alternative to the addition of protective substances is to subject the bacteria to various stress conditions during their culturing. Bacterial cells adapt after their exposure to stress factors (e.g., heat shock or osmotic stress) which can improve their stability over time and their survival during the drying process. Osmotically stressed *L. acidophilus* ATCC 4356 cells showed better shelf life during long term ambient storage after fluid bed drying in comparison with unstressed fluid bed dried cells [8]. During drying and processing, the probiotic cells are exposed to different stresses. Mechanical stress, mainly shearing forces, occurs during atomization of the microbial solution, osmotic stress is present during dehydration, and thermal stress during the drying stages. More precise analytical methods that can recognize both cellular damage and the metabolic activity of cells are a valuable tool for the production and assessment of probiotics. The viability assessment of probiotics is most commonly conducted by the plate count method—an established and trusted method that lacks, however, the recognition of non-growing cells, and cannot differentiate between the different levels of metabolic activity in the cell. While critical in measuring the number of viable cells in dried probiotic products, it tends to underestimate the cell numbers, especially because of differences in sample preparation, rehydration and dilution preparation [9]. It is also unable to recognize the viable but nonculturable cells (VBNC), which can also show beneficial probiotic effects [10]. Single cell analysis using flow cytometry with fluorescence staining can be used to establish an alternative viability assay for probiotic preparations [11]. For the simultaneous assessment of cell membrane permeability and overall cell viability, a dual staining approach with propidium iodide and RedoxSensor^TM^ Green can be applied [12]. Cell sorting was also introduced to confirm the results obtained by flow cytometry. This allows to sort a selected number of cells (from a single cell up to 10^6^) onto either a 96-well plate with liquid medium or a Petri dish with agar. Such analysis was conducted for the cells subjected to process stresses (pH shock and uncontrolled pH during culturing, heat shock during drying) to show the metabolic and physiological differences between the analyzed preparations and to demonstrate the potential of the applied method.

## 2. Results and Discussion

For cellular staining, two dyes were used: RedoxSensor^TM^ Green and propidium iodide. Each of these allows for the evaluation of different parameters. RedoxSensor^TM^ Green is used to assess the metabolic activity in cells. It is membrane permeable and binds to the reductase enzymes, forming a green fluorescent (520 nm) product. PI in turn is not membrane permeable, which allows for the assessment of membrane integrity in the analyzed cells. Damaged cells absorb the dye, which then binds with DNA and emits a red fluorescence (635 nm). Using simultaneous staining with the mentioned dyes with complex samples presented a distribution indicating four physiologically different subpopulations (Figure 1). The active group shows high values for RSG, indicating high metabolic activity, while the PI signal levels demonstrate almost no cellular damage. The second group, described as dead cells, shows exactly the opposite—high signal values for PI and almost none for RSG. There are also two additional groups, marked mid-active I and mid-active II. In these groups both RSG and PI simultaneously emit their signal, which means that cells in those groups are both damaged and metabolically active (Figure 2). Additionally, the growth of these cells after sorting into liquid and agar media is significantly hindered. Such cells suit the descriptions of VBNC (viable but non-culturable) cells and are omitted in plate cell counting on pour plates [13]. Their recognition is especially important, since due to their metabolic activity these cells could still bring potential health benefits in probiotics.

### 2.1. Viability and Activity of Treated Strains

#### 2.1.1. Control

Cells in the control samples were grown in optimal conditions. This resulted (Figure 3) in most of the cells belonging to the active subpopulation (62.9% ± 0.22 for *E. faecium*, 77.5% ± 1.20 for *L. mesenteroides* and 69.7% ± 0.35 for *C. divergens).* The dead cells accounted for no more than 10% of the total number of cells (9.15% ± 0.62 for *E. Faecium,* 8.02% ± 0.60 for *L. mesenteroides* and 8.07% ± 0.37 for *C. divergens).* The mid-active II subpopulation constituted 17.93% ± 0.26 for *E. faecium*, 6.21% ± 0.73 for *L. mesenteroides* and 15.35% ± 0.27 for *C. divergens.* The mid-active I subpopulation occurred the least, with 0.84% ± 0.07 for *E. faecium*, 0.39% ± 0.10 for *L. mesenteroides* and 0.21% ± 0.08 for *C. divergens.*

#### 2.1.2. Heat Shock

Heat shock samples presented differences between the tested strains (Figure 4). For *E. faecium* and *C. divergens*, most of the cells belonged to the active subpopulation (63.79% ± 1.12 for *E. faecium,* 25.23% ± 1.20 for *L. mesenteroides* and 57.87% ± 1.50 for *C. divergens).* Meanwhile, *L. mesenteroides* proved to be less resistant to heat stress, and the dead cells were the most numerous subpopulation for that strain (6.12% ± 0.68 for *E. faecium,* 26.5% ± 0.88 for *L. mesenteroides* and 10.50% ± 0.37 for *C. divergens).* The mid-active II subpopulation reached values of 21.60% ± 1.24 for *E. faecium,* 33.86% ± 0.18 for *L. mesenteroides* and 21.46% ± 1.17 for *C. divergens.* The mid-active I subpopulation occurred the least, with 0.09% ± 0.06 for *E. faecium,* 1.03% ± 0.17 for *L. mesenteroides* and 0.41% ± 0.14 for *C. divergens.*

#### 2.1.3. pH Shock

Cells in the control samples were exposed to a pH shock. This resulted (Figure 5) in most of the cells being dead (71.68% ± 1.23 for *E. faecium*, 83.86% ± 0.53 for *L. mesenteroides* and 65.05% ± 0.65 for *C. divergens).* The active cells accounted for no more than 10% of the total number of cells (3.17% ± 0.68 for *E. faecium*, 3.04% ± 0.12 for *L. mesenteroides* and 9.08% ± 0.26 for *C. divergens).* The mid-active I subpopulation occurred the least, with 2.06% ± 0.27 for *E. faecium*, 1.03% ± 0.10 for *L. mesenteroides* and 6.63% ± 0.91 for *C. divergens.* The mid-active II subpopulation constituted 16.54% ± 1.03 for *E. faecium*, 8.47% ± 0.56 for *L. mesenteroides* and 11.49% ± 0.76 for *C. divergens.*

#### 2.1.4. Uncontrolled pH

Cells in the control samples were grown without pH control. This resulted (Figure 6) in the active and dead cell subpopulations distributing evenly. Active: 32.22% ± 1.00 for *E. faecium*, 32.32% ± 1.85 for *L. mesenteroides* and 29.85% ± 1.39 for *C. divergens.* Dead: 34.59% ± 0.20 for *E. faecium*, 34.25% ± 2.09 for *L. mesenteroides* and 32.88% ± 1.73 for *C. divergens.* The mid-active II subpopulation was calculated at 11.08% ± 0.53 for *E. faecium*, 20.58% ± 0.52 for *L. mesenteroides* and 11.02% ± 0.73 for *C. divergens.* The mid-active I subpopulation was observed in 14.22% ± 0.53 for *E. faecium*, 6.15% ± 0.36 for *L. mesenteroides* and 17.79% ± 0.54 for *C. divergens.*

#### 2.1.5. Dried Samples

Cells in the dried samples were exposed to temperature as well as dehydration. This resulted (Figure 7) in the active and dead cell subpopulations distributing evenly. Active: 36.31% ± 0.61 for *E. faecium*, 33.23% ± 0.39 for *L. mesenteroides* and 34.78% ± 0.91 for *C. divergens.* Dead: 6.49% ± 0.31 for *E. faecium,* 10.42% ± 0.42 for *L. mesenteroides* and 9.22% ± 0.43% for *C. divergens.* The mid-active II subpopulation was calculated at 41.77% ± 0.22 for *E. faecium*, 44.61% ± 0.05 for *L. mesenteroides* and 39.45% ± 0.32 for *C. divergens.* The mid-active I subpopulation occurred the least, with 0.69% ± 0.10 for *E. faecium*, 0.41% ± 0.05 for *L. mesenteroides* and 0.68% ± 0.12 for *C. divergens.*

#### 2.1.6. Coated Samples

Cells in the coated samples were exposed to a higher temperature for twice the time of the dried samples. This resulted (Figure 8) in most of the cells being dead (73.53% ± 0.59 for *E. faecium*, 87.05% ± 0.66 for *L. mesenteroides* and 90.06% ± 0.53 for *C. divergens).* The active cells accounted for no more than 1% of the total number of cells (0.99% ± 0.12 for *E. faecium*, 0.43% ± 0.07 for *L. mesenteroides* and 0.32% ± 0.20 for *C. divergens*). The mid-active II subpopulation constituted 8.67% ± 0.46 for *E. faecium*, 1.94% ± 0.30 for *L. mesenteroides* and 1.26% ± 0.15 for *C. divergens.* The mid-active I subpopulation was observed in the following percentages 11.71% ± 0.93 for *E. faecium*, 7.63% ± 0.58 for *L. mesenteroides* and 6.18% ± 0.40 for *C. divergens.*

### 2.2. Comparison of Plate Counts and Flow Cytometric Analysis

Cells were enumerated using plate counts and during flow cytometric analysis (Table 1). While the total cell counts are comparable, the differences in the number of cells counted for the same sample using both methods come mainly from the flow cytometry taking into consideration cells from groups marked as dead and mid-active, while plate counts were able to take only growing cells, marked as active, into account. Other factors affecting the cell counts in flow cytometry are the software gates that were set to eliminate non-cellular debris and other pollutants from the analytic image. Similarly, gates based on aspect ratio and area of observed objects were set to discriminate between single cells and aggregates, which lead to the removal of cell aggregates and therefore lowered the final cell count in comparison with plate cell counts. This is especially notable in the case of strongly aggregating cells, such as *E. faecium* and *L. mesenteroides*, and is more prominent in stressed samples, since the formation of aggregates is one of the defense mechanisms employed by cells undergoing stress [14,15]. The advantage of flow cytometry is its speed (no need for incubation as in plate cell counts) and the ability of universal multiparametric analysis. Plate cell counts require a range of selective media, and no single medium or isolation protocol is viable for all probiotic strains. Reliable enumeration also requires a certain number of colonies on a plate, commonly ranging between 30–300 cfu/plate, which constitutes a relatively narrow range, and the necessity for multiple dilutions further reduces the accuracy of this method [16].

### 2.3. Cell Sorting into Titration Plates and Petri Dishes

The imaging flow cytometry supported the sorting of microbial cells from the defined subpopulations, aimed at the correlation of flow cytometric data (evaluation of the structure and physiology of the bacterial cells) with the growing potential of the defined subpopulations. The potential of this approach has been demonstrated in previous work [17]. This was performed using the cell sorter’s single-cell sorting mode. The procedure involved the sorting of single bacterial cells from the defined subpopulations into separate spots on a Petri dish containing MRS-Agar medium. The Petri dish surface was divided into four separate sections corresponding to subpopulations of active, dead, and two discrete subpopulations of mid-active cells: mid-active I and mid-active II (Table 2). Using the cell sorter, 24 single cells from each sample were dropped onto plates. The growth on the Petri dishes was observed as colony forming units and was measured after 72 h of incubation to allow the colonies to form from a single cell. The measurement of growth on the titration plates was conducted using a plate reader. An optical density of 600 nm was used to determine whether growth had occurred in wells. OD600 was first measured for sterile MRS broth and then subtracted from further measurements. In this way, the threshold of OD600 that was used to detect growth was established at 0.1. The ability of the mid-active I and mid-active II groups to grow even while undergoing some cellular damage can point to those cells entering the VBNC state. This is described as a group with lower metabolic activity, unable to grow on routine laboratory media [13], which can be observed for the mid-active I group in coated *L. mesenteroides* samples. The results of the assessment of cell viability using flow cytometry correspond with the results obtained using plate cell counts. The group discrimination established based on flow cytometric data is also confirmed by the results of cell sorting. The mid-active I and mid-active II groups are characterized by differences in metabolic activity (higher for mid-active II), and the mid-active II group was able to grow on plates, while there was little to no growth for samples from the mid-active I group. The effect of incubation time is also noteworthy: differences in growth were observed not only for different media but also after 72 h in comparison with growth after 48h.

### 2.4. Imaging Flow Cytometry and Its Use in the Assessment of Diversity within Complex Populations

The uniqueness of the conducted research lies in the prospect of the use of imaging flow cytometry (IFC) in combination with specific fluorescent staining to enable the in-depth characterization of the physiological states of bacterial cells from tested samples. This was due to gaining a higher data resolution and the direct correlation of fluorescence intensity measurement (definition of subpopulation) with cellular morphology. The correlation with cell morphology, based on the digital signal processing of generated images of the analyzed objects (cells), provided a significant step forward with regard to interpretation of the results. Thus, the simultaneous discrimination of bacterial cells from non-cellular debris, e.g., biofilm particles and single cells from aggregates, was facilitated. BacLight™ RedoxSensor™ Green Vitality Kit (Invitrogen, Thermo Fisher Scientific, Eugene, OR, USA) was used to characterize and distinguish the different physiological states of the bacterial cells, providing the definition of subpopulations of active, dead, and two discrete subpopulations of mid-active cells: mid-active I and mid-active II (Figure 9). Thus, the significant resolution of the applied assay enabled the monitoring of the physiological status of the bacterial cells, revealing functional heterogeneity of microbes within the tested samples at single cell level [18].

The in-depth characterization of the physiological states of bacterial cells from tested samples accomplished using the IFC-based assay is also associated with the concept of the supported sorting approach. This concept assumes the use of the significant resolving power of the processed IFC data to define precise boundaries (gate definition) of microbial subpopulation for cell sorting experiments. This approach implemented in our research improved the unique cellular feature definition to provide more specific isolation of microbial cells from active, dead and two discrete subpopulations of mid-active cells: mid-active I and mid-active II subpopulations. Flow cytometry allows to observe VBNC cells in commercial preparations [19,20] and can be used to provide more accurate information much faster than classical microbiological methods, which require long incubation times. Other methods used as an alternative to plate cell counts, such as fluorescent in-situ hybridization, and nucleic acid-based enumeration methods, such as reverse transcriptase PCR (RT-PCR) and real-time quantitative PCR (qPCR), are useful tools for research and clinical trials yet are not suitable, however, for quality control or industry [16,21]. Herein lies the versatility of flow cytometry, which can be used for the multiparametric analysis of cells in various applications, both for research and industrial alike.

## 3. Materials and Methods

### 3.1. Inoculum Preparation

Strains chosen for this experiment were *L. mesenteroides 51* KBiMŻ, *E. faecium 73* KBiMŻ, and *C. divergens* 3 KBiMŻ. All three strains were identified using MALDI-TOF mass spectrometry. MRS broth was chosen for propagation as a medium ensuring optimal growth conditions for lactic acid bacteria. To obtain the highest biomass yield, it is necessary to prepare inoculum in a volume of 10% of the bioreactor culture. Therefore, for a culture volume of 1 L, preparation was conducted in two stages, gradually increasing the volume, which allowed for better adaptation of microorganisms and shortened the resting phase in culture. Preparation of the inoculum was carried out in a laminar chamber to reduce the risk of infection. The first step was to thaw the strain, stored in the freezer, which was carried out on ice to limit cell damage that could occur in case of faster thawing. After the microorganisms had thawed and reached room temperature, they were transferred to a 15 mL Falcon conical tube containing 9 mL MRS broth, which was then sealed with parafilm and incubated for 24 h at 30 °C. After incubation, 10 mL of the inoculum was transferred to a flask containing 90 mL of MRS broth, followed by another 24 h incubation at 30 °C. After these steps, the inoculum was ready to be used to start a culture in the bioreactor.

### 3.2. Bacterial Cultures

Cell biomass was cultured using Biostat A plus bioreactors [Sartorius]. Before culturing, the pH electrode was calibrated against buffers at pH 4 and pH 9. One liter of MRS broth medium was used for seeding. The bioreactor was subsequently autoclaved at 121 °C with a 20 min exposure time to provide sterile conditions. After sterilization and cooling of the bioreactor, the inoculum was added in a volume of 10% of the volume of the medium (100 mL), while the bioreactor was pressurized with nitrogen to minimize the risk of infection with airborne microorganisms. In the inoculated bioreactor, the culture was grown at 30 °C, with a stirrer speed of 150 RPM and pH set at 6.5 (the optimal value for the selected strains). The pH value was monitored and kept constant by regulation with a 30% NaOH solution. The culture was grown for 24 h, and the end of the exponential growth phase was conferred by a graph of NaOH consumption and pH changes over time. The stabilization of pH at the set level with the simultaneous absence of base consumption indicated the end of microbial growth and inhibition of the production of acidifying metabolites. After completion of the culturing, the culture was pumped into sterilized centrifuge vessels with a spout hose, using a peristaltic pump.

### 3.3. Stress Factors during Culturing

To introduce stress conditions, certain parameters were changed for different cultures, e.g., short-term (30 min) thermal and acid stress were introduced by increasing the temperature (up to 50 °C) or by changing the pH to 2.5. In another variant the culturing was also conducted without pH control. The purpose of these changes in conditions was to check whether the stress induced on bacterial cells significantly affects their survival during fluidized bed drying.

### 3.4. Fluid Bed Drying

GEA Strea-1 laboratory fluid bed dryer was used for the drying process. Firstly, the matrix (crystalline microcellulose or starch products) was added to the drum of the fluid bed dryer. Next, the stream of filtered and heated air was introduced through the bottom perforated plate, allowing to keep the matrix in the fluid phase and ensuring even drying in a set temperature range (up to 50 °C). Microorganisms suspended in the solution of a protective substance (5% trehalose) were fed to the dryer by an external peristaltic pump. They entered an atomizing nozzle, supplied with air under the pressure of 2 bar. Drying and coating took approximately 30 min for each step using 100 g of matrix and after completion of the process the obtained samples were packed for storage and further analysis.

### 3.5. Plate Count Method

The plate counting method was used in combination with flow cytometry to determine the number of live microorganisms present in the samples after drying and in the bioreactor culture. This was carried out under sterile conditions under a laminar chamber, in duplicate, and the decimal dilution method was used to prepare the samples. It involves the preparation of tubes containing 9 mL of solvent (0.9% NaCl), and then pipetting 1 mL of the sample into the first tube, so that a dilution of 10^−1^ is obtained. After thoroughly vortexing the tube to mix the sample evenly, 1 mL of the sample with a dilution of 10^−1^ is transferred to the next tube, and the whole process is repeated until the required order of dilution is obtained. In the case of cell cultures, it was necessary to prepare dilutions of up to 10^−9^ for the resulting samples. For the culture fluid and suspension for drying it was possible to take 1 mL directly for dilution, however, in the case of the finished formulation, i.e., the granulate, it was necessary to rehydrate it. For that purpose, 1 g of sample was weighed into a 99 mL flask with 0.9% NaCl, which was then placed in a 37 °C water bath. After 30 min of shaking in a water bath, further dilutions were prepared using the suspension. 1 mL of diluted samples were applied to the Petri dishes. The samples on the plates were then poured with MRS-Agar medium (previously sterilized and stored at 55 °C to prevent solidification of the medium) cooled to about 45 °C, mixed thoroughly and then allowed to solidify. The Petri dishes were incubated in aerobic conditions in an incubation chamber at 30 °C for 48 h, after which time visible colonies were counted and evaluated. The results of counting visible colonies are considered statistically significant only for plates with the dilution sample for which the number of colonies ranged from 30 to 300.

### 3.6. Flow Cytometry and Cell Sorting

Flow cytometry is a cell count method that was used as an alternative method to classic plate cultures. Bacterial cells were examined for cellular metabolic activity and viability using imaging flow cytometer Amnis FlowSight™ (Luminex Corp., Austin, TX, USA) equipped with three lasers (405 nm, 488 nm and 642 nm), five fluorescence channels (acquisition by a multi-channel CCD camera), and side scatter detector (SSC). Post-acquisition data analysis was performed using the IDEAS software (Luminex Corp., Austin, TX, USA). Three-step morphological characteristics of the analyzed cells were performed: (i) in the first step the Gradient RMS parameter from brightfield signals (Ch01) was used, which enabled the discrimination of the high resolution cell images, (ii) the second step involved the brightfield digital image processing parameters: Aspect Ratio and Area to characterize shape and size of the analyzed bacterial cells in combination with the discrimination of bacterial cells from non-cellular debris (particles from prebiotic components) and single cells from aggregates. The viability and activity of probiotic bacteria cells in the samples was determined by fluorescent staining with RedoxSensor^TM^ Green (Figure 10) and PI (propidium iodide) (Figure 11). The samples for analysis were prepared by centrifugation and then suspended in a 1% PBS buffer in 1:200 dilution. Then, the following dyes were added to the samples pipetted into Eppendorf tubes in a volume of 500 ul: 1.6 µL of RedoxSensor^TM^ Green and 1.2 µL of PI. The cells in the samples were counted and assessed for morphology (microscopic image), activity (signal for RedoxSensor^TM^ Green) and integrity of the cell membrane (signal for PI). After the mentioned steps, the cells were sorted to isolate and further analyze populations of interest (e.g., cells with reduced metabolic activity—VBNC). The sorter allows to isolate a designated number of cells into titration plates with MRS broth or Petri dishes with MRS-Agar medium. Cell sorting is conducted based on readings obtained using fluorescent staining. Four different cell groups were observed in samples, namely active, mid-active I, mid-active II and dead cells. They were determined by two separate parameters—their metabolic activity (RedoxSensor^TM^ Green) and viability (PI). Cells from all groups were then sorted using the BD FACS Aria™III (Becton Dickinson, USA) cell sorter into separate spots on Petri dishes or separate wells on 96-well titration plates. The configuration of the instrument was as follows: four lasers (375 nm, 405 nm, 488 nm and 633 nm), eleven fluorescence detectors, forward scatter (FSC) and side scatter (SSC) detectors; 70 μm nozzle and 70 psi (0.483 MPa) sheath fluid pressure. For growth on Petri dishes, the colony forming units were enumerated after 48 h of incubation at 30 °C. For titration plates, the optical density at 600nm was measured in all wells after 48 and 72 h.

### 3.7. Machine-Learning Assisted Discrimination of Cells vs. Debris in Tested Samples

A machine learning (ML)-based protocol was employed to facilitate and improve the discrimination of cells from cellular and non-cellular debris (Figure 12). The Machine Learning (ML) module is incorporated in to the IDEAS^®^ 6.3. software, which was designed to process data acquired by Amnis Flow Sight imaging flow cytometer (Luminex Corp., Austin, TX, USA). After manual selection of the two “truth” populations, the ML algorithm calculated two super features (classifiers) that maximally separated each “truth” population from the others. The implementation of the ML module was described in detail in the work of Konieczny et al. [22]. For every classifier, truth populations between 31 and 32 events were manually tagged and loaded into the ML module. ML generated and tested features of all eight main categories corresponding to the BF and SSC channels. The resulting cells_vs_debris_classifier contains a series of seven differentially weighted features/parameters (Table 3). Both classifiers were plotted into histograms, and events with values higher than zero belong to images that are best represented by their classifier.

## 4. Conclusions

In this study, the effects of stress conditions on three different fluid bed dried probiotic strains were measured, combining and comparing both plate cell counts and flow cytometry. Samples analyzed by flow cytometry were then further sorted into different identified groups: active, dead, mid-active I and mid active II (with mid-active I and mid active II possibly being VBNC). These selected groups were further analyzed to compare their viability, activity, and probiotic potential, and to compare their occurrence with the plate count method. Such an approach can help to identify the importance of VBNC cells in probiotic preparations and can also prove the importance of supporting plate counts with additional, more accurate techniques such as flow cytometry for more viable and detailed probiotic cells assessment.

## Figures and Tables

**Figure 1 ijms-24-06841-f001:**
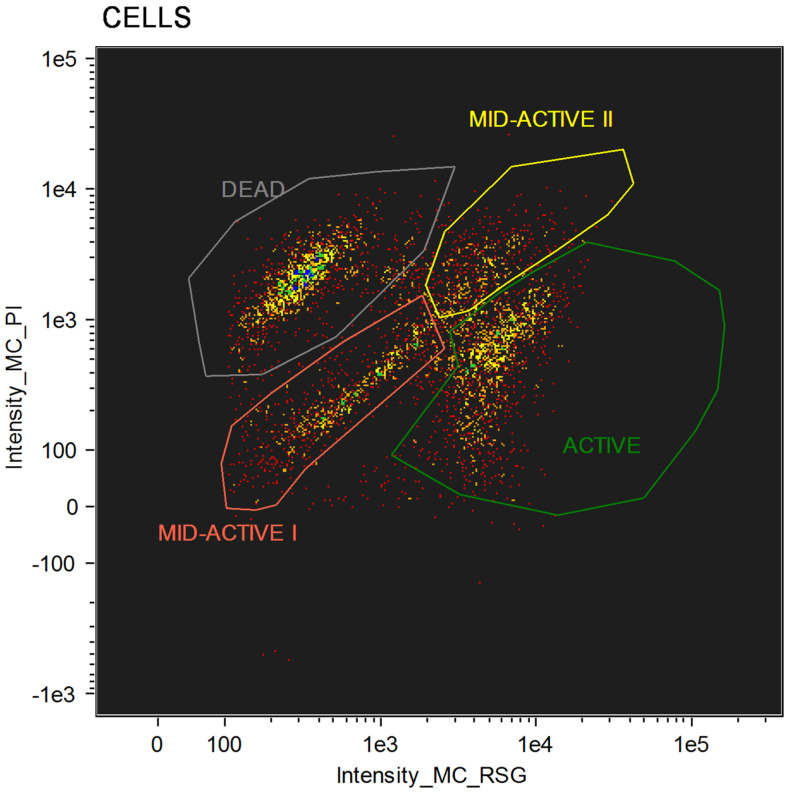
Group distribution after flow cytometry with RedoxSensor^TM^ Green and propidium iodide staining.

**Figure 2 ijms-24-06841-f002:**
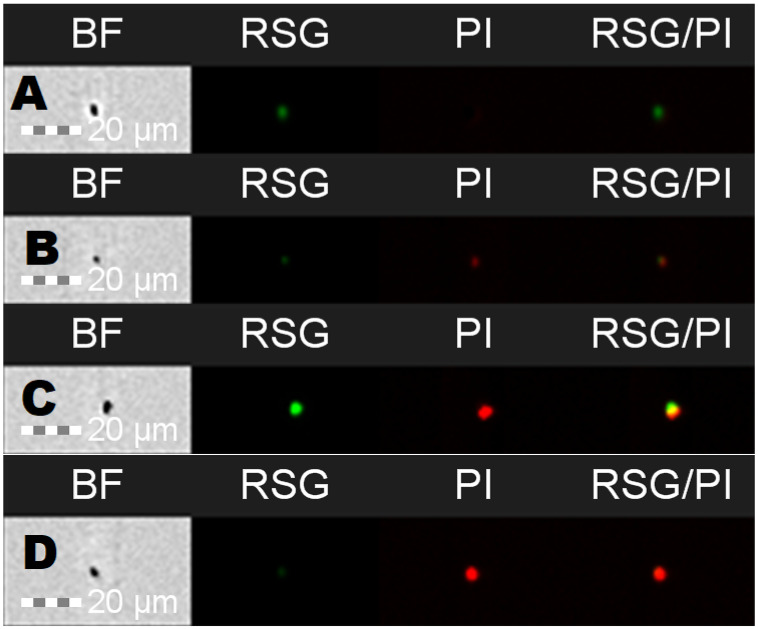
Flow cytometric images of cells for different channels: BF—brightfield, RSG—RedoxSensor^TM^ Green, PI—propidium iodide, RSG/PI—mixed signals for both dyes. Samples represent different subpopulations: A—active, B—mid-active 1, C—mid-active 2, D—dead.

**Figure 3 ijms-24-06841-f003:**
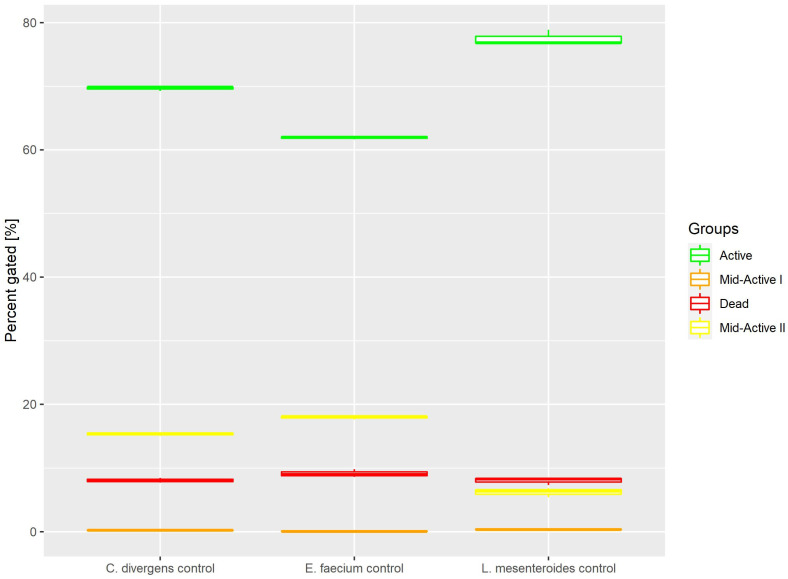
Cell subpopulation percentages in control samples. The distribution of cells between different groups was measured after culturing in optimal conditions.

**Figure 4 ijms-24-06841-f004:**
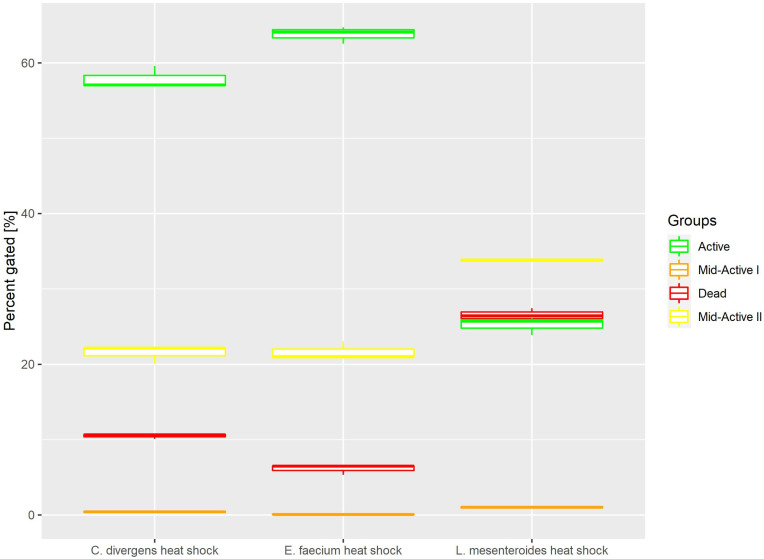
Cell subpopulation percentages in samples after heat shock. The distribution of cells between different groups was measured after incubating cells at 50 °C for 30 min.

**Figure 5 ijms-24-06841-f005:**
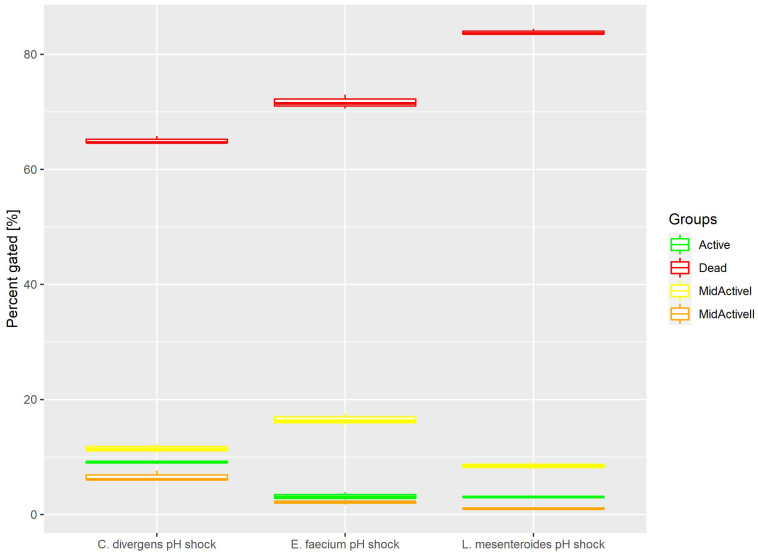
Cell subpopulation percentages in samples after pH shock. The distribution of cells between different groups was measured after exposing cells to pH 2.5 for 30 min.

**Figure 6 ijms-24-06841-f006:**
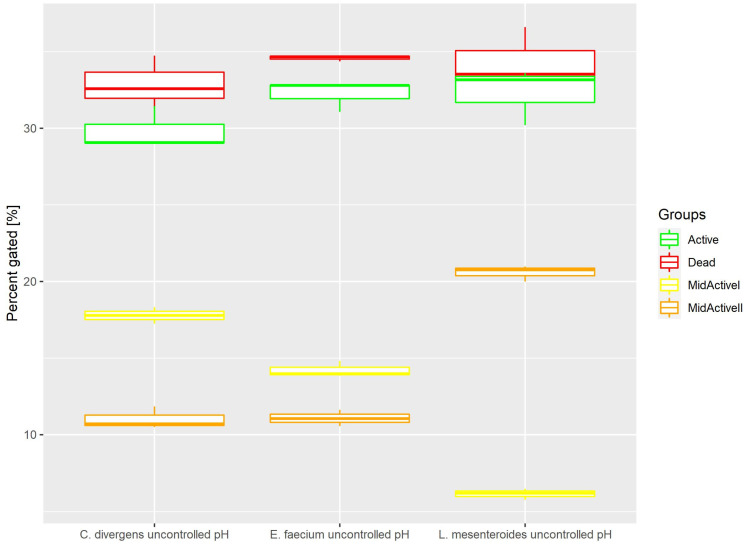
Cell subpopulation percentages in samples cultured without pH control. The distribution of cells between different groups was measured after culturing without pH regulation.

**Figure 7 ijms-24-06841-f007:**
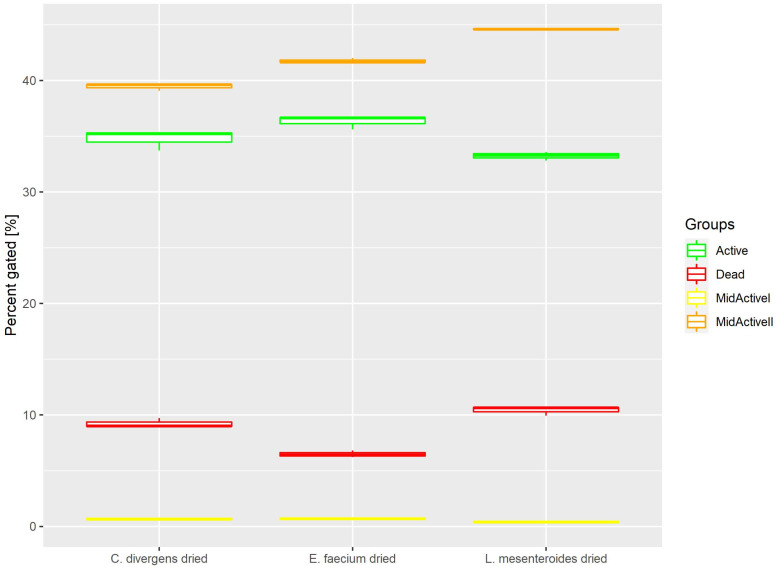
Cell subpopulation percentages in samples after fluid bed drying.

**Figure 8 ijms-24-06841-f008:**
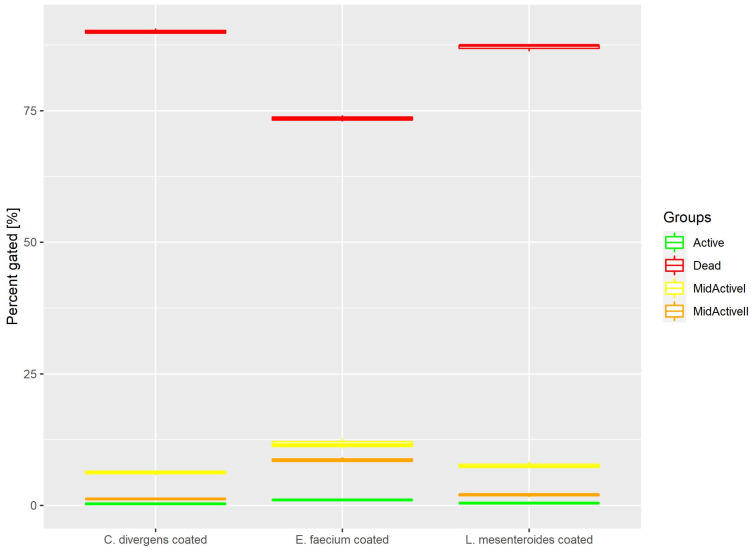
Cell subpopulation percentages in dried samples after coating.

**Figure 9 ijms-24-06841-f009:**
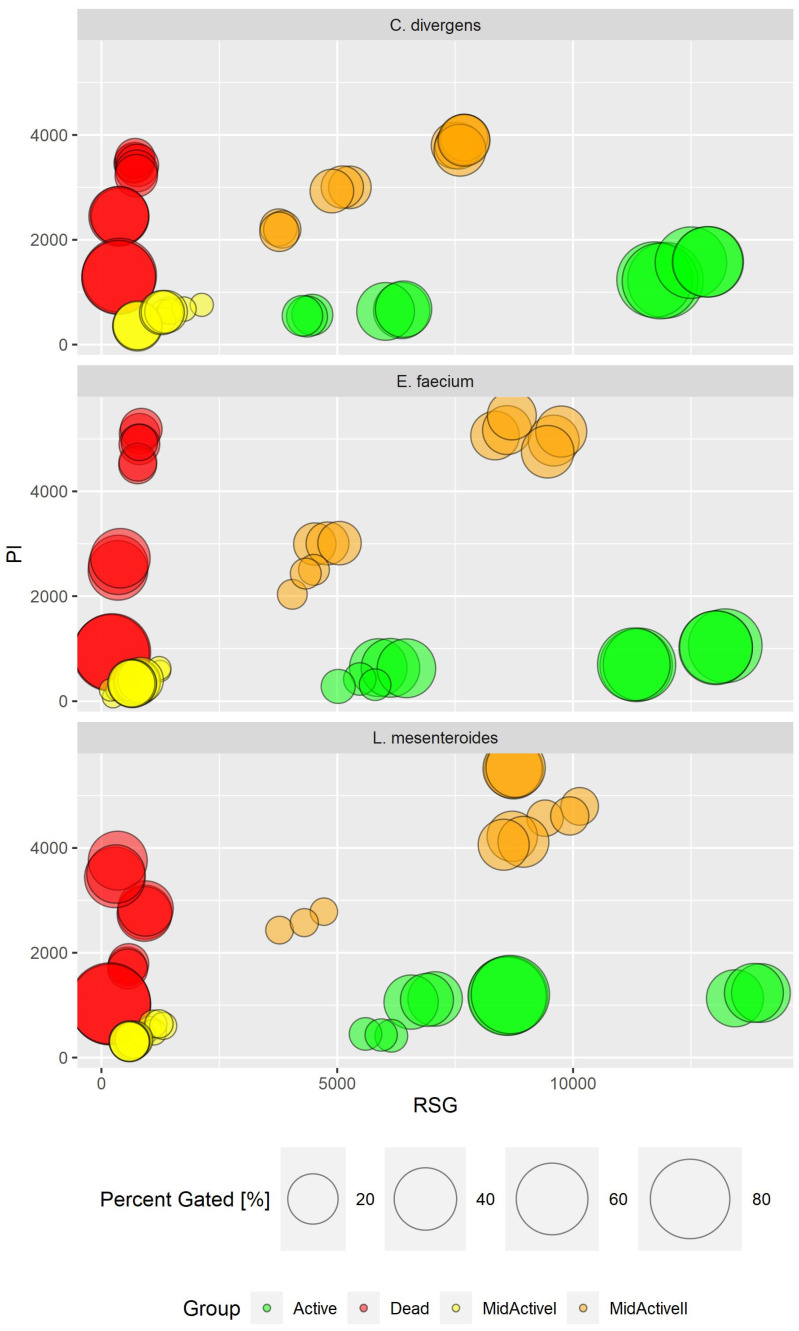
Subpopulation distribution in different strains by PI and RSG signal value (in RFU—relative fluorescence units). Comparison of results for all experiments.

**Figure 10 ijms-24-06841-f010:**
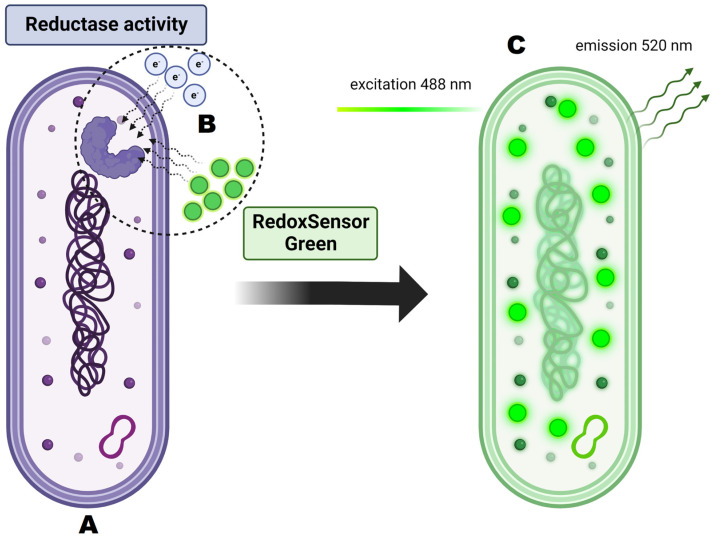
Principles of RedoxSensor^TM^ Green staining and its activity on the cellular level. (**A**) Analyzed cell is dyed with RedoxSensor^TM^ Green. (**B**) Dye molecules permeate the cell membrane and interact with reductases. Reductase activity reflects changes in electron transport chain function and in vitality. (**C**) After excitation at 488 nm the dye emits green-fluorescent signal at 520 nm.

**Figure 11 ijms-24-06841-f011:**
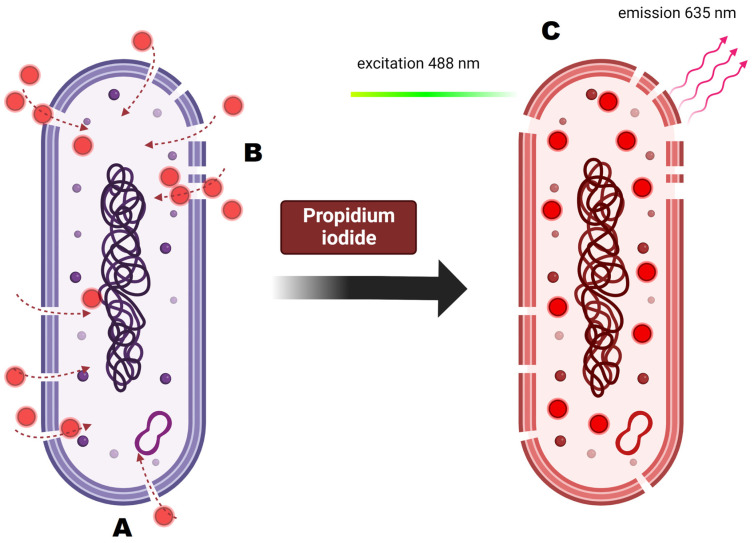
Principles of propidium iodide staining and its activity on the cellular level. (**A**) Analyzed cell is dyed with propidium iodide. (**B**) PI is membrane impermeable so it enters only the cells with damaged cell membrane. (**C**) Once inside the cell, PI binds to DNA by intercalating between the bases with no preference. After binding, the fluorescence is enhanced and excited at 488 nm, resulting in emission maximum at 635.

**Figure 12 ijms-24-06841-f012:**
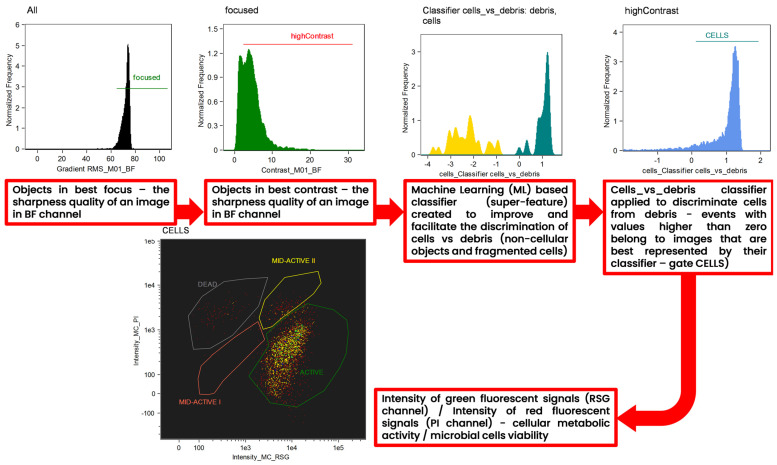
Gating strategy applied for the interpretation of the results of imaging flow cytometry (IFC) analysis. Analysis was assisted by an advanced tool to improve the interpretation of the cytometric results—the machine learning (ML) module of IDEAS software. “Batch” populations being representative images of microbial cells and debris (non-cellular objects and fragmented cells) were specified by the user to initiate a generation of the super feature (classifier) for the precise discrimination of single microbial cells vs. debris. Both classifiers were plotted into histograms, and events with values higher than zero belong to images that are best represented by their classifier.

**Table 1 ijms-24-06841-t001:** Comparison of bacterial cell enumeration using plate cell counts and flow cytometric analysis.

Sample Name	Plate Cell Counts [cfu/mL]	Flow Cytometry [obj/mL]				
		Total	Dead	Mid-Active I	Mid-Active II	Active
*L. mesenteroides*	2.7 × 10^8^	3.57 × 10^8^	2.16 × 10^7^	1.05 × 10^6^	1.68 × 10^7^	2.10 × 10^8^
*L. mesenteroides* heat shock	1.29 × 10^8^	1.85 × 10^8^	3.52 × 10^7^	1.36 × 10^6^	4.50 × 10^7^	3.35 × 10^7^
*E. faecium*	4.05 × 10^8^	2.72 × 10^8^	2.17 × 10^7^	2.02 × 10^5^	4.26 × 10^7^	1.47 × 10^8^
*E. faecium* heat shock	5.85 × 10^8^	2.53 × 10^8^	1.32 × 10^7^	2.04 × 10^5^	4.65 × 10^7^	1.37 × 10^8^
*C. divergens*	4.05 × 10^8^	3.40 × 10^8^	2.13 × 10^7^	5.68 × 10^5^	4.05 × 10^7^	1.84 × 10^8^
*C. divergens* heat shock	9.35 × 10^7^	2.50 × 10^8^	2.13 × 10^7^	8.37 × 10^5^	4.36 × 10^7^	1.18 × 10^8^

**Table 2 ijms-24-06841-t002:** Growth rates of bacterial cells after sorting on titration plates with solid medium and 96-well plates with liquid medium.

Sample Name	MRS-Agar on Petri Dishes [cfu]				MRS Broth on 96-Well Plates [OD600 > 0.1]			
	Active	Mid-Active I	Mid-Active II	Dead	Active	Mid-Active I	Mid-Active II	Dead
*E. faecium* dried 48 h/72 h	20.8%/37.5%	0%/0%	8.3%/8.3%	0%/0%	75%/83.3%	0%/0%	8.3%/16.7%	0%/0%
*E. faecium* coated 48 h/72 h	20.8%/29.2%	0%/0%	25%/29.2%	0%/0%	20.8%/29.2%	0%/0%	12.5%/12.5%	0%/0%
*L. mesenteroides* dried 48 h/72 h	4.2%/8.3%	0%/0%	8.3%/8.3%	0%/0%	4.2%/8.3%	0%/0%	0%/0%	0%/0%
*L. mesenteroides* coated 48 h/72 h	20.8%/33.3%	0%/0%	4.2%/12.5%	0%/0%	8.3%/20.8%	4.2%/12.5%	4.2%/4.2%	0%/4.2%

**Table 3 ijms-24-06841-t003:** Components of the cells_vs_debris classifier—parameters and weights indicating the discrimination efficiency.

Parameters	Weights
H Entropy Std_M09_Ch09_9	−18.39
H Entropy Std_M01_BF_9	−17.57
H Homogeneity Std_M01_BF_11	−12.9
Area_MC	−12.89
H Entropy Std_M01_BF_11	−12.88
H Entropy Mean_M09_Ch09_9	−12.69
Major Axis Intensity_M09_Ch09	−12.68

## Data Availability

Data is contained within the article.

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
