# Peer review of "Imaging Flow Cytometry Demonstrates Physiological and Morphological Diversity within Treated Probiotic Bacteria Groups"

_ijms, 2023, doi:10.3390/ijms24076841_

Round 1
Reviewer 1 Report
In this work, the authors have elucidated microbial diversity in the probiotic group using flow cytometry. By introducing flow cytometry in addition, to the conventional plate cell counts method, the authors were able to identify and measure the effects of stress conditions on 3 different fluid-bed-dried probiotic strains. The samples analyzed by flow cytometry were then further sorted into different identified groups: active, dead, mid-active I, and II. Following the sorting, these selected groups were further analyzed to compare their viability, activity, and probiotic potential and to compare their occurrence with the plate count method. This approach allowed the authors to identify the importance of VBNC cells in probiotic preparations and prove the importance of flow cytometry in addition to plate counts for probiotic cell assessment.
Overall, the work is good, but the figures need work. I’m outlining my comments below:
1. Figure 1 must be explained clearly with a figure legend.
2. Figure 2 must be explained clearly with a figure legend.
3. Figure 5 should be replaced with a better-quality image. Y-axis does not have a label. Figure 5 must be explained clearly with a figure legend.
4. Figure 6 should be replaced with a better-quality image. Y-axis does not have a label. Figure 6 must be explained clearly with a figure legend.
5. Figure 7 should be replaced with a better-quality image. Y-axis does not have a label. Figure 7 must be explained clearly with a figure legend.
6. Figure 8 should be replaced with a better-quality image. Y-axis does not have a label. Figure 8 must be explained clearly with a figure legend.
7. Figure 9 should be replaced with a better-quality image. Y-axis does not have a label. Figure 9 must be explained clearly with a figure legend.
8. Figure 10 should be replaced with a better-quality image. Y-axis does not have a label. Figure 10 must be explained clearly with a figure legend.
9. Figure 11 is pixelated. The authors should replace it with better quality image.
10. Line 34: There is an extra space between the words “the” and “introduction” and “of” and “more”. Please eliminate these 2 extra spaces.
Reviewer 2 Report
The authors demonstrated that imaging flow cytometry can be used to count the bacteria in a probiotic formulation, evaluating their physical and morphological characteristic. They exploit the metabolic activity using a specific dye (RedoxSensorTM green) and the membrane damage using the propidium iodide. To confirm their findings, they sorted the different groups to assess the growth. They found that probiotic bacteria are composed by four different groups (active, dead, mid-active I and mid-active II).
The manuscript is well written, and it is very useful in probiotics field. Although I have several concerns that need to be addresses before publication.
MAJOR
· The inoculum preparation was performed in a “laminar chamber” as stated by the authors in lines 87-88 and line 134. Did they manipulate the bacteria in an anaerobic condition? Why did they not use a bugbox anaerobic workstation? Please explain this choice.
· The authors should state where the petri dishes were incubated. In anaerobic or in aerobic condition?
· The authors use the imaging flow cytometry and in the abstract, they explain the advantage to use this technology compare to the classical flow cytometry, but along the manuscript this is not well described. The tests that they did using the IFC are also reproducible using the classical flow cytometry. They should highlight this concept along the manuscript.
· The title comprises “flow cytometry” and not “imaging flow cytometry”. They should modify accordingly.
· The authors did not show the gating strategy. This should be included at least as supplementary data.
MINOR
· Line 345 it is present twice “If”. The authors should fix it.
· Line 408 the authors should correct the word “be6ying”.
Round 2
Reviewer 2 Report
The authors replied to all my comments.
In my opinion, the manuscript can be accepted for publication in the present form